# Decline of Fine Suspended Sediments in the Madeira River Basin (2003–2017)

**Irma Ayes Rivera** [1,*], **Elisa Armijos Cardenas** [2], **Raúl Espinoza-Villar** [2], **Jhan Carlo Espinoza** [2,3], **Jorge Molina-Carpio** [4], **José Max Ayala** [5], **Omar Gutierrez-Cori** [2,6], **Jean-Michel Martinez** [7] and **Naziano Filizola** [1,8]

1    Postgraduation Program CLIAMB, Instituto Nacional de Pesquisas da Amazônia (INPA)—Universidade do Estado do Amazonas (UEA), Ave. André Araújo, 2936, Manaus CEP 69060-001, Brazil; nazianofilizola@ufam.edu.br

2    Instituto Geofísico del Perú (IGP), Calle Badajoz 169, Urb. Mayorazgo IV etapa, Ate, Lima 15012, Peru; earmijos@igp.gob.pe (E.A.C.); respinozavillar@gmail.com (R.E.-V.); jcev09@gmail.com (J.C.E.); omar.gutierrez@lmd.jussieu.fr (O.G.-C.)

3    Univ. Grenoble Alpes, IRD, CNRS, Grenoble INP, Insitut des Géosciences de l'Environnement (IGE, UMR 5001), 38000 Grenoble, France

4    Instituto de Hidráulica e Hidrología (IHH), Universidad Mayor de San Andrés, Casilla 699, Campus Universitario, Calle 30 Cota Cota, La Paz 15000, Bolivia; amolina@umsa.bo

5    Instituto Hondureño de Ciencias de la Tierra (IHCIT), Universidad Nacional Autónoma de Honduras (UNAH), Ciudad Universitaria, Boulevard Suyapa, Tegucigalpa 11101, Honduras; jmax_ayala@hotmail.com

6    Laboratorie de Météorologie Dynamique (LMD), Institut Pierre Simon Laplace (IPSL), Sorbone Université, Univ Paris 06, 75252 Paris, France

7    Géosciences Environnement Toulouse—GET (Centre national de la Recherche Scientifique—CNRS, Institut de Recherche pour le Développement—IRD, Université Paul Sabatier—UPS), Observatoire Midi-Pyrénées—OMP, 14 rue Edouard Belin, 31400 Toulouse, France; martinez@ird.fr

8    Departamento de Geociências, Universidade Federal do Amazonas (UFAM), Ave. General Rodrigo Otávio, Jordão Ramos 6200, Campus Universitário, Coroado I, Manaus CEP 69077-000, Brazil

*    Correspondence: ayesrivera@hotmail.com; Tel.: +55-92-984-686-783

**Abstract:** The Madeira River is the second largest Amazon tributary, contributing up to 50% of the Amazon River's sediment load. The Madeira has significant hydropower potential, which has started to be used by the Madeira Hydroelectric Complex (MHC), with two large dams along the middle stretch of the river. In this study, fine suspended sediment concentration (FSC) data were assessed downstream of the MHC at the Porto Velho gauging station and at the outlet of each tributary (Beni and Mamoré Rivers, upstream from the MHC), from 2003 to 2017. When comparing the pre-MHC (2003–2008) and post-MHC (2015–2017) periods, a 36% decrease in FSC was observed in the Beni River during the peak months of sediment load (December–March). At Porto Velho, a reduction of 30% was found, which responds to the Upper Madeira Basin and hydroelectric regulation. Concerning water discharge, no significant change occurred, indicating that a lower peak FSC cannot be explained by changes in the peak discharge months. However, lower FSCs are associated with a downward break in the overall time series registered at the outlet of the major sediment supplier—the Beni River—during 2010.

**Keywords:** Madeira River; fine suspended sediment concentrations; water discharge; hydroelectric dams

## 1. Introduction

The fluvial suspended sediment dynamic plays an essential role in transporting nutrients, maintaining water bodies' stability, and supporting ecological habitats. The interactions of multiple natural and anthropogenic factors acting on different temporal and spatial scales in a watershed can control or modify the erosion rate, allowing sediment transport or promoting deposition and resuspension processes in the river channel [1]. As observed worldwide by Walling [2] and in the Andean Amazon by Aalto et al. [3] and by Pepin et al. [4], suspended sediment loads are good indicators of large-scale climatic variability and anthropogenic activities [5–7].

A major challenge for holistic sediment transport studies is the highly complex interaction between the hydrological, geological, and climatic variables which influence the sediment transport processes. In many cases, a reductionist analysis is performed. Given this framework, this study focuses on the Amazon River Basin, and particularly on the Madeira River. The Amazon is the world's largest river in terms of water discharge (210,000 $m^3 \cdot s^{-1}$) [8], and the main supplier of suspended sediment load to the Atlantic Ocean (600–1100 $Mt \cdot year^{-1}$) [9,10]. The Madeira River, the second largest Amazon tributary in terms of water discharge and the largest contributor to sediment load (~50%), is formed by the confluence of the Beni and Mamoré Rivers (Figure 1), which originate in the Andean Mountains (the main sediment source) [11–13]. The geomorphologic characteristics (i.e., the Andean Mountains, floodplains, and Brazilian Shield) are key factors in the sediment transport process.

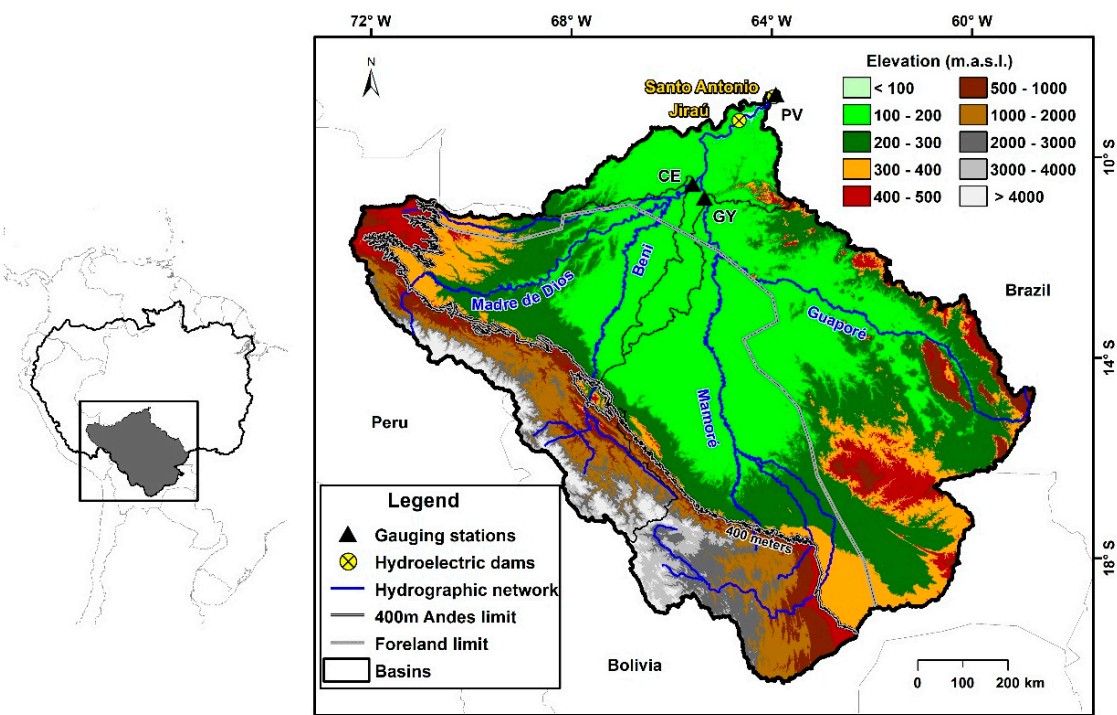

**Figure 1.** Upper Madeira River Basin with three gauging stations: Porto Velho (PV) in the Madeira River, Cachuela Esperanza (CE) in the Beni River, and Guayamerín (GY) in the Mamoré River. The Madeira Hydroelectric Complex (MHC) is formed by the Jirau and Santo Antonio dams. Elevation data were obtained from the Shuttle Rada Topography Mission (SRTM) [14].

The complex tropical climate over the Amazon–Andean region [15–17] also plays a fundamental role in suspended sediment load and its variability. The climate variability in the Madeira Basin and the entire Amazon region has drawn the attention of the scientific community due to the occurrence of extreme hydro-climatic events [16,18–21]. This larger climate variability may support a more efficient production and transportation of sediments [22]. For instance, as analyzed by Espinoza et al. [23], the abrupt transition from an extreme drought year (2010) to the highest discharges in the Peruvian

Amazon (Ucayali and Marañón Rivers) in 2011 generated the lowest and highest suspended sediment concentrations ever recorded, respectively. Hence, an improvement in the understanding of how the climate affects fluvial sediment transport can provide useful information when assessing other possible influences of sediment processes, such as human activities.

Among these activities, many developing countries use hydraulic energy as their main source of energy, and the Upper Amazon Basin has been a focal point given its high hydroelectric potential [24,25]. However, previous studies [24–29] documented ecological and economic damages occurring, including risks to food security, before and after dam operations. Indeed, in the Amazon region, a major economic and food source is fishing. The potential hydraulic energy has already been partly used by the Madeira Hydroelectric Complex (MHC), with an installed capacity of nearly 7000 Mw (i.e., 11% of the Brazilian hydroelectric installed capacity) [30]. There is, however, concern regarding the direct and indirect effects of hydroelectric projects. Worldwide, the decrease in sediment transport downstream of dams is a major effect [31–34] due to the impact dams have on the river channel and floodplain geomorphology [24], floodplain productivity [35], and ecosystem stability [25,28]. Latrubesse et al. [24] categorized the Amazon sub-basins according to their vulnerability to cumulative impacts from existing and proposed hydroelectric projects. The Madeira Basin was assigned with the largest dam environmental vulnerability index (DEVI), which indicates the potential risk of hydrophysical and biotic disturbance in the Amazon floodplains related to the construction of hydroelectric projects [24].

As the Madeira River has a significant impact on the Amazon sediment balance, a robust and cost-efficient monitoring system of fluvial sediments is required for the Madeira River and its Andean tributaries. In addition, attaining an understanding of the relevant processes that affect sediment connectivity is a major scientific challenge. In order to achieve the latter, the aims of this study were: (1) to analyze the spatio-temporal evolution of the fine suspended sediments over the Madeira River for the 2003–2017 period, considering hydrological information from downstream and upstream of the MHC in the main Andean tributaries, and (2) to examine the fine suspended sediment regime in the pre- and post-MHC. These results can be used to support further management decisions in the Madeira River.

## 2. Materials and Methods

### 2.1. Study Area

The Madeira Basin is situated between 4.5° and 20° S and 56.5° and 73.5° W, and extends over an area of $1.4 \times 10^6$ km$^2$, which represents 23% of the Amazon Basin ($6 \times 10^6$ km$^2$). The basin has an elevation ranging from 25 m a.s.l. in the Brazilian lowlands to 6400 m a.s.l. in the Andean Cordillera. The Madeira River is formed by the junction of the Beni and Mamoré Rivers (Figure 1). At its confluence with the Amazon, the Madeira River delivers an estimated mean annual discharge of 31,200 m$^3 \cdot$s$^{-1}$ [36] of white water (i.e., accounting for low organic and high inorganic material levels, based on Wallace's water classification (1853), which describes the water color features according to the organic/inorganic materials in the water as described by Egerton [37]), comprising 16% of the Amazon's discharge. Along the middle stretch, the Madeira River has two large hydroelectric projects in Brazil: Santo Antonio and Jirau (Figure 1). Both were identified as possible suitable sites in 1971 by the Brazilian Mine and Energy Ministry, were constructed from 2008 to 2012, and named the MHC [38]. Santo Antonio's dam is located 6.0 km upstream from the Porto Velho (PV) gauging station. At PV, which measures the Upper Madeira Basin (Figure 1), a mean annual discharge of 18,500 m$^3 \cdot$s$^{-1}$ was estimated for the 1967–2017 period. Previous studies observed negative trends for the monthly minimum and annual mean discharges for the Upper Madeira from 1974 to 2004 [39], and a negative trend for the monthly minimum discharge from 1967 to 2013 [16]. These trends can be related to an increase in droughts across South America and the Amazon region [16,40], and an increase in the dry-season length in the Bolivian Amazon [41,42].

The contribution of the Madeira Basin at PV is considered in this study as the Upper Madeira Basin. Of its total area, 11% is situated in Peru, 73% in Bolivia, and 16% in Brazil [16]. Among its geological features and steep topography, highly erodible rocks promote the incision of channels, rapid mass washing, and high sediment production [11]. Latrubesse and Restrepo [43] estimated that the Bolivian Andes have the highest sediment yield in the Cordillera at $>2070$ t·km$^{-2}$·year$^{-1}$. As a result, the Madeira Basin supplies around 500–600 Mt of suspended sediment into the Amazon River per year [13,44].

River gradients across the forelands (i.e., regions of potential sediment accumulation formed by the Andes' contraction and the craton in response to geodynamic processes [45]) are only a few tens of centimeters per kilometer [44]. The grain size of suspended sediments in the Madeira River varies between 0.02 and 0.10 mm (i.e., fine sand, silts, and clay) [46], representing approximately 94% of the Madeira River solid load [47]. At the downstream end of the floodplain, tectonic and hydraulic control is clearly identified where the Beni and Mamoré Rivers reach the Brazilian Shield near the Brazilian–Bolivian border. The two outlet gauge stations from the Bolivian National Meteorological and Hydrological Service (SENAMHI in Spanish) are located over this part of the basin. These are the Cachuela Esperanza (CE) station for the Beni River, which is approximately 130 km after its confluence with the Madre de Dios River, and the Guayaramerin (GY) station for the Mamoré River, which is approximately 190 km downstream of its confluence with the Guaporé River.

The Mamoré River, even though it originates in the Andes, the slope break at the piedmont and entering the plains, reduces the energy available in the river channel to transport sediment from the Andean tributaries [13]. This strongly influences sedimentation processes in the Andean Piedmont [13,44]. The Guaporé River, on the other hand, originates from the Brazilian Shield and has 66% of its basin in it. This is an area characterized by its high long-term geomorphic stability and very slow denudation rates (0.02 mm·year$^{-1}$) [48,49]. This causes the black water of the Guaporé River (i.e., accounts for high organic and low inorganic materials), contributing to a low suspended sediment concentration (SSC) downstream.

In terms of basin rainfall, just over half of the accumulated annual rainfall occurs between December to March [15,16]. However, in the Madeira Basin, the rainfall amount ranges widely from ~300 mm·year$^{-1}$ in the mountains to ~600 mm·year$^{-1}$ in the eastern Andean foothills [17,50]. The estimated mean annual rainfall at CE is 1800 mm·year$^{-1}$, and at GY is 1600 mm·year$^{-1}$ [16]. As documented by previous studies, the mean annual discharge peak at GY generally occurs in mid-April, and around the end of February and the beginning of March at CE. The Madeira River's flow at Porto Velho (where the discharge peak is observed at the beginning of April) is controlled by the hydrological regimes of both tributaries—the Beni and Mamoré Rivers.

*2.2. Data and Methodology*

The temporal hydrological variability was assessed on a monthly time scale for the Madeira hydrological year, from September to December (year$_{t-1}$) to January to August (year$_{t0}$). For instance, the 2014 flood is said to have occurred in the hydrological year from September 2013 to August 2014 (i.e., each hydrological year has its wet and dry season). On a sub-basin scale, for the gauging stations at the outlet of the two sub-basins (CE and GY), discharge (Q) time series from 1983 to 2017 were provided by the SENAMHI (Figure 1 and Table 1). Additionally, PV discharge data from 1967 to 2017 were provided by the National Water Agency of Brazil (ANA in Portuguese).

The suspended sediment cross-section (i.e., average concentration along a river section) includes a fine suspended sediment fraction ($>0.45$ and $<63$ μm) that is mostly related to clay and silt material. It also contains a coarse fraction ($>63$ μm) related to sand. As observed by Espinoza-Villar et al. [22], the fine fraction dominates the surface samples (99%) in the Madeira main stem. Because of this, and because a homogeneous vertical gradient was found at the Amazon main stem for the fine suspended sediment fraction [51], we focused only on the fine suspended sediment concentration (FSC).

Suspended sediment concentrations, collected three times per month at the surface (through the SO-HYBAM Observatory), are available online for PV, CE, and GY for the period 2003–2017 [52]. For more information on the sediment concentration measurement procedure and its data quality, refer to the research by Vauchel et al. [13]. When in situ observed data were missing (February–March 2016), satellite data from the Moderate Resolution Imaging Spectroradiometer (MODIS) sensors onboard the Aqua and Terra satellites were used to obtain reflectance information at PV. These data were converted into surface suspended sediment concentrations based on the method by Espinoza-Villar et al. [22]. Surface suspended sediment measurements were interpolated to generate daily time series through the Hydraccess program [53], which is freely available through the SO-HYBAM Observatory website. Finally, to obtain the average FSC over the water column, Armijos et al. [54] found that FSCs are related to surface concentrations by a factor of 1.17.

**Table 1.** Gauging station characteristics for 2003–2017.

| River-Gauging Station Code | Latitude (degree) | Longitude (degree) | Altitude (m) | Drainage Basin (km$^2$) | Q Max (m$^3 \cdot$s$^{-1}$) | Q Mean (m$^3 \cdot$s$^{-1}$) | Q Min (m$^3 \cdot$s$^{-1}$) | Mean Surface Suspended Sediment Concentration (mg$\cdot$L$^{-1}$) |
|---|---|---|---|---|---|---|---|---|
| Beni and Madre de Dios—Cachuela Esperanza (CE) | −10.5374 [1] | −65.5846 [1] | 110 [1] | 280,900 [1] | 32,300 [2] | 9150 [2] | 1800 [2] | 660 [2] |
| Mamoré and Guaporé—Guayaramerin (GY) | −10.8124 [1] | −65.3430 [1] | 120 [1] | 611,700 [1] | 29,800 [2] | 7100 [2] | 660 [2] | 200 [2] |
| Madeira—Porto Velho (PV) | −8.7700 [1] | −63.9104 [1] | 43 [3] | 976,000 [3] | 59,000 [3] | 18,500 [3] | 2080 [3] | 360 [3] |

Source of data: [1] Vauchel et al. [13], [2] SENAMHI [55], and [3] ANA [56]. Q Max is the maximum period discharge, Q Mean is the 2003–2017 annual mean discharge, and Q Min is the minimum period discharge.

The discharge and FSC time series were analyzed in monthly and interannual time steps, considering the suspended sediment load peak (December to March) and evaluating possible relationships using regression analysis techniques. This was performed for the time series and for mean anomalies (i.e., monthly value subtracted by its mean). To evaluate the homogeneity, changes, and temporal trends for the 2003–2017 period, the following statistical tests were applied to the monthly and FSC peaks in the discharge and FSC time series: (1) the nonparametric Pettitt test [57] to seek change points along the time series, estimating the *p*-values using Monte Carlo resampling; (2) the Buishand test [58] and Von Neumann ratio to evaluate the homogeneity of the time series; (3) the Mann–Kendall test [59,60] to evaluate temporal trends over the time series; and (4) flow duration curves to assess the exceeded discharge probability with the surface suspended sediment measurements for hydrological years. Outcomes with 90 and 95% confidence intervals (CIs) are shown in the results section.

Possible changes in the mean FSC due to the hydroelectric facilities were evaluated through the Student's *t*-test for pre-dam and post-dam periods. The Santo Antonio and Jirau reservoirs started to fill during September 2011 and 2013, respectively. However, it was not until 2015 that MHC operated as designed (i.e., each dam generating with all 50 turbines) [61,62]. Between the reservoirs filling and the start of operations, the largest flood registered at PV occurred in the austral summer of 2014 [18]. As such, the post-MHC period was considered to be from September 2014 to August 2017.

## 3. Results

### 3.1. Q and FSC Variability

At CE in the Beni River, the monthly discharge began to increase in October, with a peak in February, and from then until May, it reduced. The same behavior was observed for the FSC (Figure 2a). In the case of GY in the Mamoré River, the discharge began to increase from November, with a peak

in mid-April and a reduction that extended until July. However, in terms of FSC, the peak occurred between January and February, two months earlier than the discharge (Figure 2b). At PV, the discharge started to increase from October, with its peak in March, and the FSC peak occurred a month before, in February (Figure 2c). A large variability was observed during the hydrograph rising limb and peak for both, the discharge and FSC at CE. Discharge was, however, more variable from the peak to the falling limb at GY and PV. For FSC at GY, the largest variability was observed during the rising limb. The minimum and maximum points, shown in Figure 2, represent drought and floods events, respectively. For instance, during the 2005 drought that affected the central and southern Amazon Basin, this generated discharges below the 25th quartile at CE from January to May, and the extreme low point in April and May at PV (Figure 2a,c). Compared with the 2014 flood event that affected a large portion of the Madeira Basin, extreme high values were observed at the three stations. In some cases, these were twice the value of the mean, as in April at GY (Figure 2b). These extreme discharges were accompanied by high FSC at CE and PV (75th quartile) during 2005, and by low concentrations in 2014. The extreme FSC low values at CE and PV were observed during February and March 2014 (Figure 2a,c).

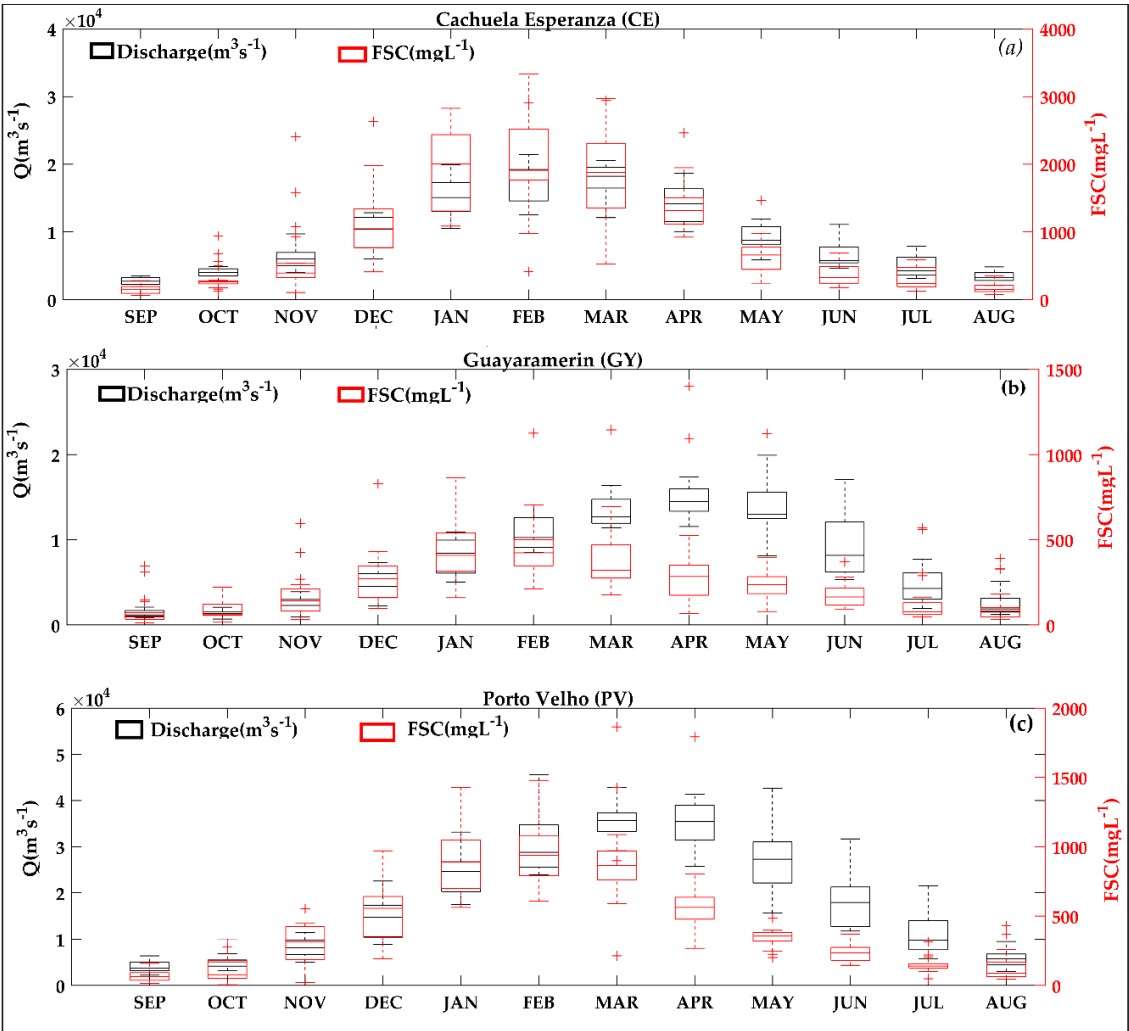

**Figure 2.** Monthly discharge (Q) in $m^3 \cdot s^{-1}$ and monthly fine suspended sediments (FSC) in $mg \cdot L^{-1}$ (**a**) at Cachuela Esperanza (CE) in the Beni River, (**b**) at Guyaramerin (GY) in the Mamoré River, (**c**) and at Porto Velho (PV) in the Madeira River. Note that the y-axes have different ranges according to the station.

The analysis of the flow duration curves (FDC) at PV provides more information on the discharge and FSC relationship during low and high flows. The discharge that was exceeded by 50% in 2005 was 12,850 $m^3 \cdot s^{-1}$. Compare this with the 2014 flood, which was 26,000 $m^3 \cdot s^{-1}$ (twice that of 2005). When considering FDC with the instant SSC measured, we observed that a high SSC concentration was measured during February 2005 (13,000 $mg \cdot L^{-1}$) (Figure 3b). This was in contrast with the low SSC measured in February 2014 (500 $mg \cdot L^{-1}$) (Figure 3c). While plotting the yearly FDC, 2014 and 2015 are highlighted. Both years have a similar 50% exceeded discharge rate, but the highest discharges (>43,000 $m^3 \cdot s^{-1}$) were observed from February until May in 2014. In terms of SSC, despite 2015 having similar discharges as 2014, the SSC was 700 $mg \cdot L^{-1}$ in February and increased to 1200 $mg \cdot L^{-1}$ in March (Figure 3d).

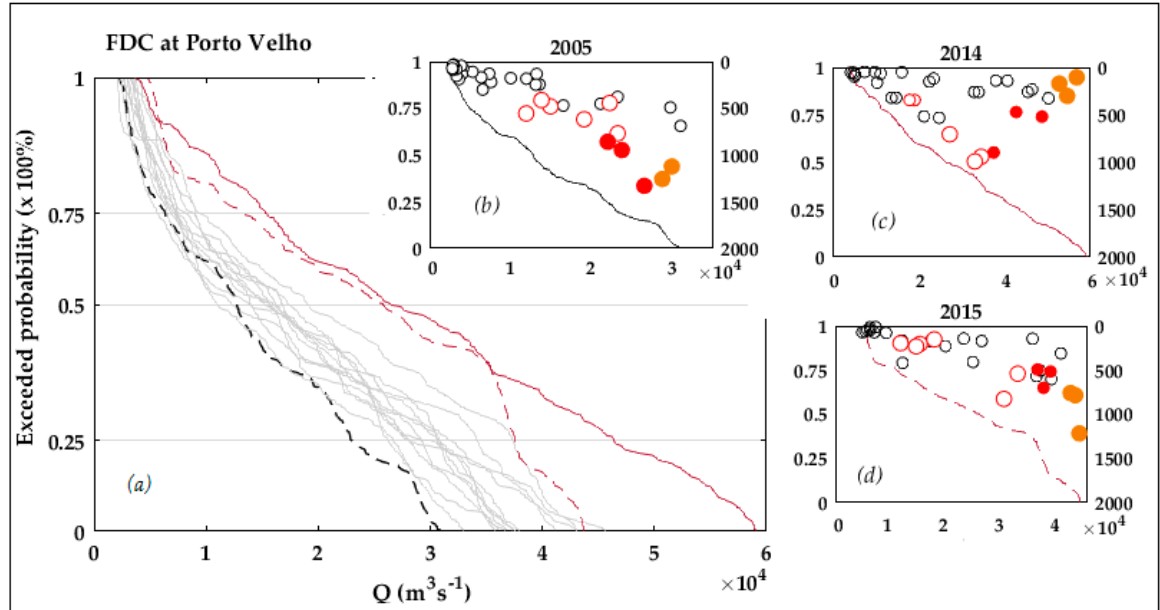

**Figure 3.** (**a**) Flow duration curves (FDC) for the hydrological years 2003–2017 at Porto Velho (PV) in the Madeira River. The red line indicates the flood event from 2013 to 2014. The black line indicates the 2005 drought and the red dashed line represents 2014–2015. FDC: (**b**) for the 2005 drought within each measured surface suspended sediment (SSC) by SO-HYBAM as empty black circles, (**c**) for the 2014 flood event, and (**d**) for 2015. December and January are represented with a red border. February is represented by red circles. March is represented in orange. For the inside figures, the left y-axis shows the exceeded probability in percentages, and the right y-axis shows the SSC in $mg \cdot L^{-1}$.

## 3.2. Madeira Hydroelectric Complex (MHC) Influences

The mean FSC at CE in the Beni River is significantly correlated with the FSC at PV in the Madeira River for the 2003–2017 period ($R^2$ = 0.36; $n$ = 181; $p$ < 0.0001; Figure 4a). This does not occur at GY ($R^2$ = 0.02; $n$ = 181; $p$ < 0.0001; Figure 4b) Then, a pre- and post-dam period analysis was completed for the CE and PV gauging stations. When considering the post-MHC period (2015–2017), a 36% decrease occurred in the mean FSC during the sediment peak (from December to March –DJFM) at CE ($p$ = 0.005), compared to the mean FSC of 2003–2008. The decrease in FSC propagated downstream such that, at PV, it was observed to be 30% less ($p$ < 0.0001). The discharge during the sediment peak did not show a significant decrease at CE (11% less, $p$ = 0.30), nor at PV (4% less, $p$ = 0.37), when compared to the previous periods.

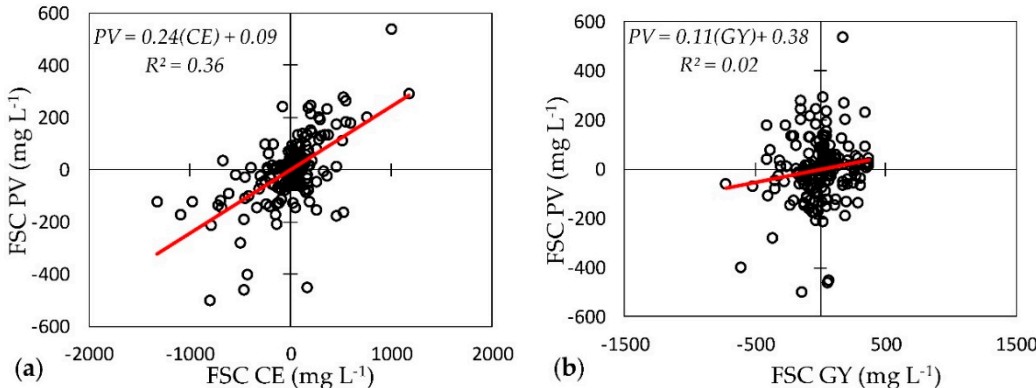

**Figure 4.** The 2003–2017 mean anomalies for the fine suspended sediment concentrations. (**a**) CE in the Beni River, related to PV in the Madeira River; (**b**) GY in the Mamoré River, related to PV.

### 3.3. Trends and Changes in Q and FSC Time Series

For the 2003–2017 period, no significant trend was found for the discharge at the CE and PV gauging stations, while a negative trend was observed for the FSC only at PV (90% confidence). At CE and PV, a significant downward break (90%) was found using the Buishand test in 2010. In the case of the interannual concentrations during the sediment peak, significant negative trends were observed at CE and at PV at 90% and 95% CIs, respectively. These trends were accompanied by a downward change in 2010 (95%) at both stations (Figure 5a,b and Table 2).

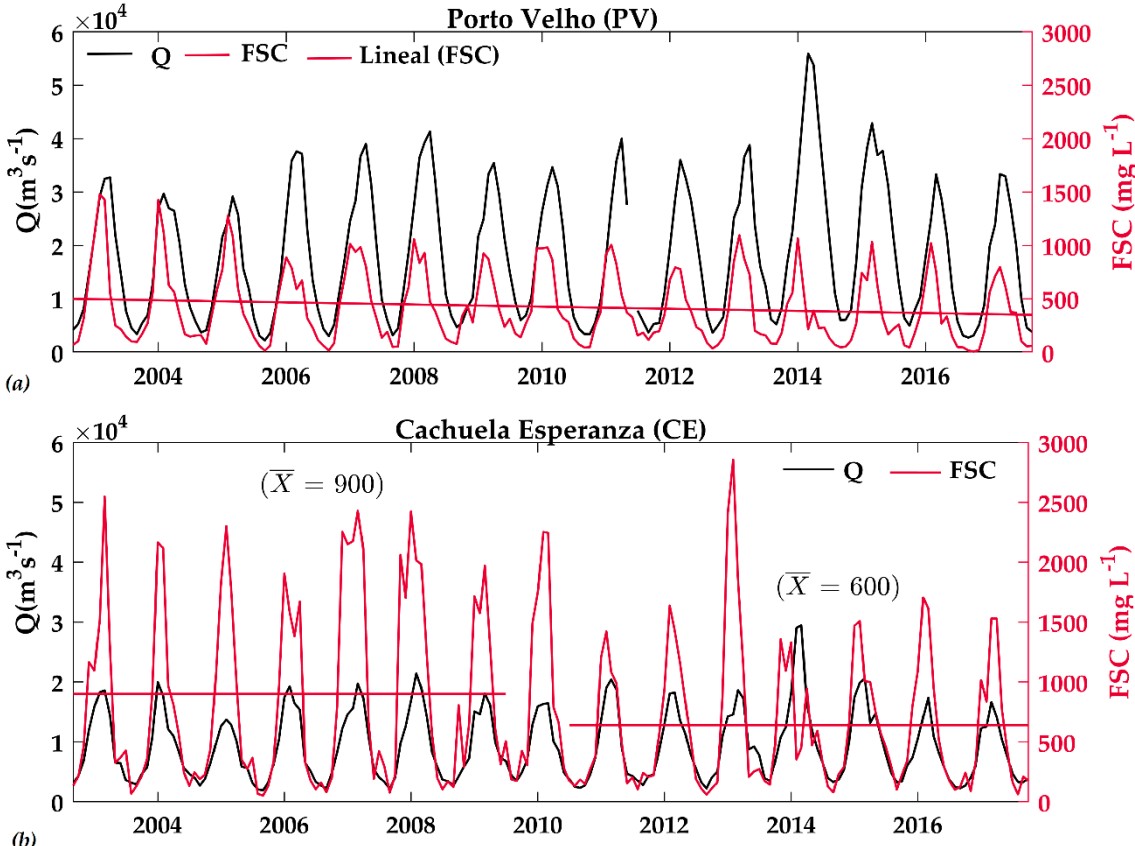

**Figure 5.** Monthly time series (2003–2017) for Q in $m^3 \cdot s^{-1}$ and FSC in $mg \cdot L^{-1}$ at (**a**) PV in the Madeira River and (**b**) CE in the Beni River. $\bar{x}$ indicates the mean FSC for two periods at CE. The second period, from January to February 2013, was not considered.

**Table 2.** Homogeneity and trend test for the monthly and sediment peak (DJFM) time series 2003–2017. *p*-values for the Pettit test (I), Buishand (II), and von Neumann (III) homogeneity tests. Kendall's τ for Mann–Kendall (MK) trend test and its associated *p*-value. Black highlight values denote significance at the 90 and 95% CIs.

| Time Series | | Q-Homogeneity Tests | | | Q-MK | | FSC-Homogeneity Tests | | | FSC-MK | |
|---|---|---|---|---|---|---|---|---|---|---|---|
| | | I | II | III | MK' τ | *p*-value | I | II | III | MK' τ | *p*-value |
| Monthly | CE | 0.262 | 0.812 | <0.0001 | 0.015 | 0.772 | 0.541 | **0.071** [1] | <0.0001 | −0.059 | 0.236 |
| | GY | 0.775 | 0.325 | <0.0001 | 0.057 | 0.258 | 0.320 | **0.098** [2] | <0.0001 | 0.063 | 0.207 |
| | PV | 0.796 | 0.314 | <0.0001 | 0.052 | 0.301 | 0.236 | **0.072** [1] | <0.0001 | −0.106 | **0.035** |
| Sediment peak | CE | 0.911 | 0.639 | 0.257 | −0.181 | 0.379 | **0.039** [1] | **0.030** [1] | 0.295 | −0.371 | 0.059 |
| | GY | 0.660 | 0.883 | 0.329 | −0.048 | 0.846 | 0.666 | 0.368 | 0.786 | 0.029 | 0.923 |
| | PV | 0.737 | 0.857 | 0.188 | −0.048 | 0.846 | **0.035** [1] | **0.023** [1] | **0.015** | −**0.600** | **0.001** |

[1] 2010 break and [2] 2009 break.

Given the decreasing sediment trend found in the FSC, a further disaggregation was done by considering the same analysis for each of the month time series. For the discharge, no significant trend nor change were found. However, for the 2003–2017 FSC, a decreasing trend was observed from December to February ($p \leq 0.074$) at Porto Velho, which was accompanied by a downward break in the mean FSC from December to February 2010 (Buishand $p \leq 0.085$; Figure 6a). At CE, a significant trend was found in December and January ($p \leq 0.074$), with the break in December and February 2010 (Buishand $p \leq 0.085$). The absence of a trend and break in the discharge series is associated with the moderate discharge–sediment relationship in DJFM, and an alteration of it during 2010 at CE ($R^2 = 0.42$, $n = 32$, and $p < 0.0001$ for 2003–2010 and $R^2 = 0.60$, $n = 28$, and $p < 0.0001$ for 2010–2017; Figure 6b,c). Note that the maximum and minimum FSC values were observed during 2013 and 2014, respectively.

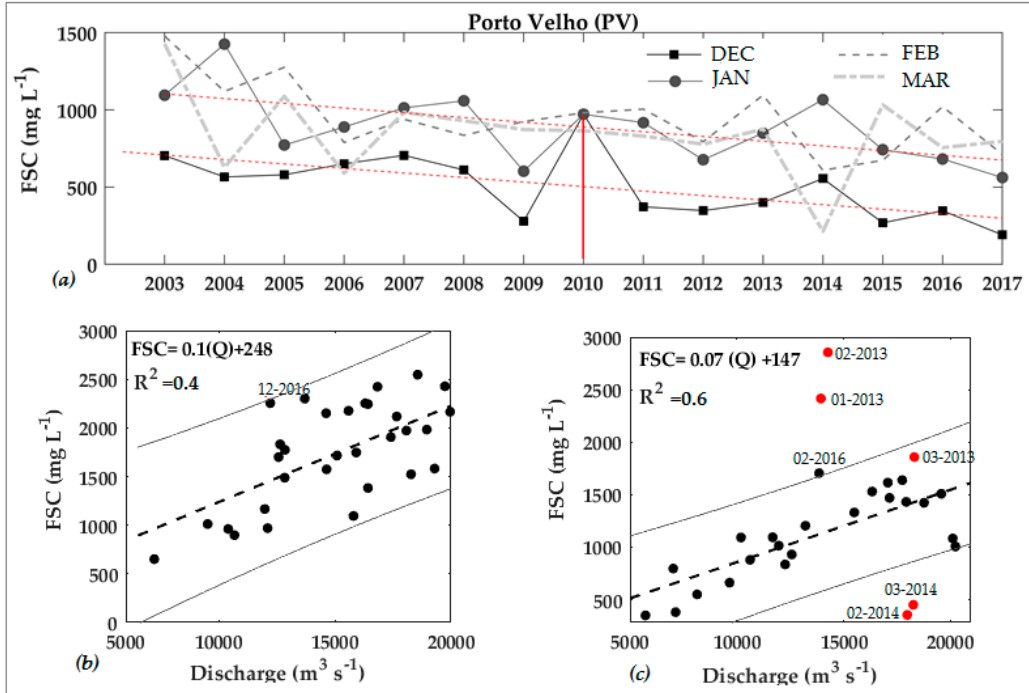

**Figure 6.** (**a**) The monthly FSC time series at PV in the Madeira River from December to March (only the December and January series with significant trends have their linear trends graphically represented as red dashed lines). The vertical line in 2010 separates the period of change in the average FSC during December and January. Water discharge and FSC during DJFM at CE in the Beni River (**b**) for 2003–2010 and (**c**) for 2010–2017 (without JF 2013 and FM 2014). The dashed black line represents the linear relationship. Black lines represent the 95% CI.

## 4. Discussion

The significant downward change in the mean FSC at Porto Velho, when comparing the pre- and post-MHC period, can be first interpreted as a direct impact from the MHC. Recently, the impacts caused by the MHC have become a major concern in the scientific community [24–29], and a reduction in the mean surface suspended sediment concentrations was found at PV between 2014 and 2015 [24]. However, the resulting negative trend and downward break upstream of the MHC suggests that the reduction in FSC at PV cannot be fully attributed to the dam's construction and operation. Moreover, an impact of the MHC on the Beni River is unlikely. Vauchel [63] evaluated the water level slope (20 cm·km$^{-1}$) between the Beni–Mamoré confluence and the Madeira River 50 km downstream of the confluence and 200 km upstream of Jirau (Nova Vida community) during the 2014 event, and concluded that the slope was maintained after the dam's construction.

These results evidence a break in the FSC at CE and PV that requires further assessment. Contrary to the FSC results, the discharge time series did not significantly change at either station for the 2003–2017 period. This could be due to a wide range of possible causes, from changes in the spatial and temporal rainfall regime to an increase in the sedimentation processes. For instance, Gautier et al. [64] observed an increase in the sedimentation rates between Rurrenabaque (i.e., Andean Beni piedmont gauging station) and CE for the 2002–2006 period. However, the lack of sediment concentrations data and granulometric characteristics in the Upper Madre de Dios Basin made it impossible to compare with the Upper Beni Basin (i.e., sediment sources). It is also important to note that considering historical discharge data over a long-term period (e.g., 1967–2013), a decreasing trend in PV has been found in relation to rainfall diminution [16,39] and increasing dry season length over the Bolivian Amazon [41].

As mentioned by Timpe and Kaplan [27], many uncertainties are associated with using short-term records to analyze the effect of hydroelectric projects on river hydrology. At the Amazonian southwestern lowlands, with discharges of more than 10,000 m$^3$·s$^{-1}$ (such as the Madeira mean discharge at Porto Velho), the same authors estimated that data collected over five years or more are required to be within 10% of the long-term mean water discharge to characterize the flow regime with 90% confidence. As such, the current results should be considered with caution, not only in terms of the water discharge changes downstream of the dams, but also in terms of suspended sediment concentrations.

## 5. Final Comments

We assessed the fine suspended sediment concentration (FSC) in the Madeira River Basin for the 2003–2017 period. We used FSC observations downstream of the Madeira Hydroelectric Complex (MHC), at Porto Velho station on the Madeira River, and upstream of the MHC, at the stations of Cachuela Esperanza (on the Beni River) and Guayaramerin (on the Mamoré River). We found a December–March FSC depletion of almost 30% ($p < 0.0001$) at Porto Velho when comparing the post- (2015–2017) with the pre-MHC (2003–2008) period. This is related to an FSC reduction of 36% ($p = 0.005$) in the Beni River (at Cachuela Esperanza), the major suspended sediment tributary of the Madeira River. The interannual evolution of fine suspended sediment concentrations between December and March at PV shows a break in 2010, which coincides with the break at Cachuela Esperanza in 2010. Water discharge time series do not show significant trends nor changes at either gauging station for the 2003–2017 period. Note that the period of analysis is relatively short for such a complex basin, during which several droughts occurred [21,23], as well as the most extreme flood event at Porto Velho since 1967. This climatic variability affected the sediment concentrations; high FSCs during relatively low discharge periods were observed (i.e., in 2005) and low FSCs were observed during the 2013–2014 flood at Porto Velho.

Though we were not able to conclude with confidence a possible explanation for the FSC changes observed in 2010 at Cachuela Esperanza, we highlight the importance of studying hydroelectric impacts based on downstream effects as well as sediment inflow from the tributaries upstream of the MHC. Future work is necessary to explore the possible causes for the FSC downward break

found in Cachuela Esperanza at the Beni River. Future dams' impact studies should also consider total suspended sediments (i.e., fine and coarse sediment concentrations), for which the mixture of water through the dam turbines and its spillway should be addressed to better understand sediment transport and resuspension.

Finally, attributing the proportional reduction in the Madeira River due to the Beni River vs. MHC remains a challenge, further complicated by the nature of the MHC as a cascade project. Assessment of sediment trapping by the MHC would require evaluating the effects from each dam, its impacts, and how they interact (i.e., if the impacts are additive or synergistic). This was not within the scope of this study and a future research can address this limitation. The impact assessment of potential changes in management and land use was not within the scope, but these factors potentially affect the observed reduction in discharge and fine suspended sediment concentration over the sub-basins.

**Author Contributions:** Conceptualization—I.A.R., E.A.C., R.E.-V., and N.F. Methodology—I.A.R., E.A.C., and R.E.-V. Software was managed by O.G.-C. Validation—I.A.R., E.A.C., R.E.-V. Formal analysis—I.A.R., E.A.C., and R.E.-V. Investigation—I.A.R., R.E.-V., E.A.C., J.C.E., J.M.-C., and J.M.A. Data curation—I.A.R., R.E.-V., and E.A.C. Original draft preparation—I.A.R., R.E.-V., and E.A.C. Reviewing and editing—I.A.R., R.E.-V., E.A.C., J.C.E., J.M.-C., J.M.A., and J.-M.M. Visualization—I.A.R., E.A.C., R.E.-V., and O.G.-C. Supervision—R.E.-V., E.A.C., and N.F.

**Funding:** The first author was supported by a Doctoral Scholarship from the Coordination for Improvement of Higher Education Personnel (CAPES) of the Brazilian Ministry of Education. J.C.E. was partially supported by the French AMANECER-MOPGA project funded by ANR and IRD (ref. ANR-18-MPGA-0008).

**Acknowledgments:** We express our gratitude to the SENAMHI from Bolivia, ANA from Brazil, and SO-HYBAM from Institut de Recherche pour le Développement (IRD) for providing the hydrological and sediment data used in this study, and the Peruvian Geophysics Institute for the internship the first author received. We would also like to thank Pascal Fraizy and Philippe Vauchel from IRD in France, Antonio Manzi from INPE in Brazil, Waldo Lavado from SENAMHI in Peru, Albert Kettner from the University of Colorado in the USA, Ana Callau Poduje from Leibniz Universität Hannover in Germany, and two anonymous reviewers for their effort and support.

**Conflicts of Interest:** The authors declare no conflict of interest. The founding sponsors had no role in the design, analysis, and interpretation of data, in the writing of the manuscript, or in the decision to publish the results.

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
