# Peer review of "Decline of Fine Suspended Sediments in the Madeira River Basin (2003–2017)"

_water, doi:10.3390/w11030514_

Round 1
Reviewer 1 Report
Review of Manuscript: Decline of fine suspended sediments for the Madeira River (2003-2017)
Authors: Irma Ayes Rivera, Raúl Espinoza-Villar, Elisa Armijos Cardenas, Jhan Carlo Espinoza, JorgeMolina-Carpio, José Max Ayala, Omar Gutierrez, Jean-Michel Martinez, Naziano Filizola
General Comments:
The objective of the paper, to assess the fine suspended sediment concentrations (FSC) of the Madeira River and its two major tributaries, is of interest to the hydro-sedimentology and water resource management communities. The results are intriguing and highlight the importance of the use of all available data to understand the impacts of anthropogenic activities, namely dam building, on riverine sediment transport. Of particular interest is the apparent observation that decreases in a major tributary sediment delivery also contributed to decreases in FSC that have been previously attributed solely to the Madeira Hydroelectric Complex. Notably, I think the conclusions of the cited Latrubesse et al. 2017 paper should be mentioned to highlight that the story in that analysis may not be complete without considering the trends of sediment loads of upstream tributaries shown here.
Overall the writing is understandable, however the sentence and paragraph structures are fair to poor and detract from the scientific message the authors wish to convey. Therefore, it is critical the article undergo a thorough revision and copy-editing before publication. Editing should focus on making the writing clearer and more concise to ensure that everything included in the manuscript is critical to the clarity or understanding of the research. There are numerous grammatical errors and sentence/paragraph structure issues throughout (e.g. few minor spelling errors, long and confusing sentences, missing transitions, word order issues, etc.). After the abstract, I stopped highlighting such errors because there were so many; I suggest having someone review the document with a fine-comb for detailed editing. Although the information gets across, the numerous errors makes the article hard to follow.
The paper presents promising results as a call for more in-depth study of the Madeira system to better understand the combined effects of the dams and upstream tributary trends in sediment load. This also serves as a broader call for developers elsewhere. However, the manuscript needs major editing for grammar and structure, and a few core concepts and the main points need highlighting throughout. I believe the manuscript is acceptable for publication with MAJOR REVISION after the authors address the following comments.
Specific Comments:
Title Suggestion: The paper presents the FSC for the Madeira, Mamore, and Beni, I might suggest including this somehow in the title, below are some suggestions:
“Decline of Fine Suspended Sediments of the Madeira River and its Major Tributaries (2003-2017)”
Or
“Decline of Fine Suspended Sediments of the Upper Madeira Watershed (2003-2017)”
In general, the title could also use the inclusion of the MHC perhaps.
Abstract:
Pg1, line 25:is originated by àis formed by the confluence of the Beni and Mamore Rivers, both originating in the Andes Mountains.
Pg. 1, line 26: rewrite sentence, structure is awkward. Perhaps – “Complex and distinguishable climate and geomorphic features of these watersheds are expressed through the hydro-sedimentological variability at the outlet of each river” (and downstream of the confluence?)
Pg 1 line 27: Rewrite. Perhaps: “The Beni River contributes more sediment to the Madeira River than the Mamore.” Sentence after this you are talking about the whole watershed (or at least mainstem Madeira) all of a sudden, I would like a transition between these two.
Pg 1: line 28,Seems to me that the main purpose of the paper is to highlight that reductions of FSC measured downstream of the MHC are not entirely related to the dams, but also a result of significant decreases in FSC in the Beni River. For this reason, I might mention the MHC earlier in the abstract, and maybe even in the title. Also, you make no mention in the Abstract that there is no downward trend in flow at both locations, which is key to your conclusion. All that said, the abstract might read better if you start with Madeira River hydro-sedimentological complexity, the dams, then the importance of the tributaries’ variability, especially the Beni. It should also be mentioned in the abstract perhaps that the tributary confluence is far upstream of the MHC reservoir.
Pg 1, line 31, do you need this sentence? Seems awkward here, again missing connection of thoughts.
Pg. 1 – line 32: Why is the 2015-2017 the post-dam period, you describe in the “ 2.2 Data and methodology” section two post-dam periods, one after the 2014 flood, and another after the flooding of Santo Antonio, choose one, and justify. If you take out the 2014 flooding for pre- and post-dam comparisons, should you take out the drought year as well? Justify.
Pg.1 – line 33 founded àfound. Larger à peak
Pg 1. Line 34 – delete alland from 2008 to.
Pg. 1 – line 36 – this sentence is confusing, it needs to be reworded for clarity.
Pg 1 line 38 – bed-channel àbedload
Editing comments stopped here, see above general comments.
Introduction
Line 42-48, great to introduce these concepts (e.g. river continuum and holistic approach), but again, hard to follow because of sentence structure issues.
Line 52 – if you are going to mention the Ganges River, it’d be nice to see some numbers, i.e. estimated percent of total loads to oceans from each.
Lines 54-55 – is this relevant to the overall story of this paper, if so, perhaps it would be worth including a mention of the association of nutrients to sediments (i.e. those that are transported as particulates. What about habitat structure provided by sediment pulses?
Line – 57 this is the first mention of the Beni and Mamore Rivers, I might describe before this that they are tributaries. Referring the reader to Figure 1 would be helpful.
Line 61 – its not complex justbecause of the Andean geology, but because it also includes the depositional piedmont, and Brazilian shield geology. Referring the reader to Figure 1 would be helpful.
Line – 75-76 – This index does not explore the explicit effects of dams onhydro-physical factors and deforestation, but rather the vulnerability of a basin to dams in the context of these other measures of vulnerability. It is a cumulative vulnerability index.
Materials and methods
Line 95 – Are you talking about monthly low and mean discharges for the 1974-2004 period as well? Specify the difference between the low discharges and monthly minimums for clarity.
Figure 1: Would be good to see an elevation key.
Pg 3 Line 109 – can we say ‘throughout the Late Cenozoic’ if the cited studies were only done in the last few decades? We are in the late Cenozoic, and more specifically the Holocene, but calling on a geological era, may not be appropriate in this study, especially since these changes are not happening over a geologic time scale (>1000s yrs).
Line 110 – define or show in figure 1 the forelands
Line 111- When are<0.5mm grains transported, during peak flows? Was this calculated from first principles for max flows in Guyot et al 1999, or is this based solely on grain size distributions of the different loads and bed material? The way this sentence is worded, it sounds like coarse sands would never be transported, even as bedload during the highest flows, is this true? Please clarify.
Line 111-112 – Are you referring to suspended, bedload, or both?
Line 129 – do you mean monthly peak discharge?
Line 136- not a very strong sentence to end with.
Line 138-139 – I don’t understand the time period you are using for analysis, do these time steps (yeart0and yeart-1) line up with wet and dry seasons, are you comparing monthly variability between two periods? Please clarify.
Lines 144-156–Reference 46 is a personal communication from one of the co-authors on this paper, but why not refer to Vauchel et al 2017 where there is mention of an analysis of uncertainty of the surface to total suspended sediment concentration relationship? I would like to see a 95% CI bounds (as calculated in Vauchel 2017) in figure 2 and the analysis.The factor applied here implies a SSCaverage = 1.17*(SSCsurface)1power law relationship as presented by Vauchel et al 2017. I would like to see more detail behind the uncertainty in moving from surface FSC to FSC that has been published. The Vauchel et al paper mentions the difficulty and infrequency of sampling at high flows; how much uncertainty do those pre-2010 peaks have? This uncertainty should be interpreted throughout the paper to show confidence in your findings.
Line 160 – specify wet season
Line 161-167- these analyses are for the entire series, not excluding the dry season, correct? please clarify.
Line 169 – 174 – Pick one of these periods and be more explicit about which will be used throughout. Also, please better justify leaving out the extreme year. Why not include the 2012-2013 water year? What explains the 2012-2013 water year FSC peak at CE?
Okay, I just saw that you corrected this year’s data, I don’t like this correction, this outlier may be an important one that the Vauchel et al 2017 uncertainty bounds would catch, that might shine light on the pre-2010 peaks, please justify correction. You do mention leaving it out for further analysis, but correcting it, seems like a stretch.
Results and discussions
Line 188 – single outlier year correct?
Line 192-193 and Figure 2a – would the 2004-2005 drought year also be considered to have a maximum FSC month along with 2002 and 2003? It might be neater to clump these events together in the writing.
Figure 2. What explains the 2012-2013 water year FSC peak at CE?
Lines 206-209 – what should I get out of this that I cannot get out of Figure 3? How should I interpret Figure 3? Why did you correct the outlier when there is such a “weak discharge-sediments relationship during DJFM” – from line 259. How does this change youre results throughout?
Figure 3 – referring to Figure 6 that doesn’t exist, should this be figure 5?
Lines 216-227 and Figure 4 – This is a very nice way to show that indeed the station at Abuna is affected by the dam, even though this far upstream is considered out of the “Direct Area of Impact” in the EIA as stated in Molina et al. Although I would try to connect this better to what follows somehow, that is, you are missing a strong transition.
Lines 237-243 – this needs to be highlighted with the lack of downward trends in flow presented in the following section, to really drive home the point.
Line 255 –significant negativetrends?
Line 260-261 – This sentence needs to be more explicit, combining what you found in the previous section to this, and bringing in the relationship between sediment load, sediment concentration, and discharge. That is, lower peak FSC cannot be explained by changes in peak discharge months, thus the downward trend in the FSC (and downward break of the overall timeseries mean) at CE can be implied to be associated with lower sediment loads.
Perhaps a fix would be to make the results and discussion sections separate?
Figure 5 label years on figure a and b.
Final Commentscovers all the important issues with the analysis. I could use a reminder of the main points related to the comments above from Line 260-261.
Author Response
The response are in the file attached.

Reviewer 2 Report
This paper presents an original study on the effects of hydrological events and dam construction on the concentration of fine sediments in the Madeira River and its tributaries. The dataset on which the study is based is very valuable. That said, the paper suffers overall from lack of clarity in presenting the results and from a feeble message.
To this reviewer's opinion, the paper need to be reformulated in order to clarify and to sharpen its message. In the following, some suggestions are provided to improve the paper in this sense.
INTRODUCTION. Some references on the topic of this paper must be included to contextualize this work and to point out what is the novelty of the study. Examples:
https://doi.org/10.3390/w10070888
https://doi.org/10.3390/w10121759
https://doi.org/10.3390/w10101302
https://doi.org/10.3390/w10101451
https://doi.org/10.1016/j.ijsrc.2017.10.003
https://doi.org/10.1016/j.geomorph.2005.09.017
http://hdl.handle.net/10459.1/59497
RESULTS AND DISCUSSION. This section is very hard to read because the figures are badly organized with respect to the text. Also, some figures are poorly described.
FINAL COMMENTS. There should be a CONCLUSION section where the main insigths of the study are highlighted.
DETAILED REVIEW
L71. Energy is measured in Mwh instead of Mw.
L108. What is the percentage of the suspended load with respect to the total load in this river?
L120. Which are the geomorphological basin characteristics of the Marmoré River?. Why don't these characteristics allow the fine sediment transport?
L125. How much are these slow erosion rates?.
L125. What is the meaning of white and black water?. Is this related to the suspended sediment concentration?. Please, clarify.
L134. Mwh instead of Mw.
The format of the tables and figures can be improved. The green background should be eliminated.
L181. Where are the FSC values of 6 - 4300 mg L-1 and 1-2300 mg L-1 shown?.
L191-200. The message of this paragraph is not clear. It must be reformulated to highlight the idea that droughts are related to high FSC and floods are related with low FSC.
L198-200. Are these data shown in any figure?.
L206-209. The values referred here do not correspond to those shown in the figures.
Figure 3. The results shown in this figure are analyzed later in the text. Reformulate the text so that the description of each figure is near the figure.
Figure 4. Are the data shown in Figure 4 relevant for this study?. If yes, make a proper description of the figure and what can be inferred from it.
L234. It is Figure 5
L237-243. Does this paragraph refer to Figure 3?. If yes, refer to that figure and put it closer.
L252. The text here refers to Figure 2a, doesn't it?.
L275-277. Do you mean that sediments are impounded and deposited?.
L278. It is Figure 6
What did happen in 2010?. Could you provide an explanation for the changes observed in 2010?
Author Response
We would like to thank the reviewer for carefully reading the manuscript and the helpful comments. Please find attached our responses. The English of the new version of the text has been revised by the Water English editors.

Round 2
Reviewer 1 Report
Review of Manuscript Revision: Decline of fine suspended sediments for the Madeira River (2003-2017)
Authors: Irma Ayes Rivera, Raúl Espinoza-Villar, Elisa Armijos Cardenas, Jhan Carlo Espinoza, JorgeMolina-Carpio, José Max Ayala, Omar Gutierrez, Jean-Michel Martinez, Naziano Filizola
There remains much room for improvement in the writing. The copy-editing and revision by the Water English editors was insufficient.The abstract and introduction particularly need further revision for clarity. While the content of the manuscript remains very important, it is still hard to follow because of the writing. I provide some specific comments below to help motivate the authors toward more clarity in presenting this important work. I again suggest having a native English speaker, who is also trained in the field of hydro-sedimentology, to review and help revise the manuscript.
Ln 27: “The complex and distinguishable…” this first sentence is unclear and does not help set the context.
Ln 29: “Both rivers originate”
Ln 30: “exploited” has a negative connotation beyond its meaning as “used”
Ln 32: “…downstream of the MHC…”
Lns 36-37: “…this responds to the…” is unclear as written.
Ln 37: “a lack of reduction” is unclear
Lns 39-40: How do the authors attribute the proportional reduction due to Beni delivery vs. MHC?
Ln 45: Remove “As part of natural river systems”
Ln 47: remove river from “sediment river loads”
Ln 48: Is the sediment connectivity systems an accepted and widely used term? What does field mean in the “field-catchment-river sediment transfer”? This section is unclear.
Ln 50: What is “It”?
Ln 52: Change to “The Amazon River watershed is the world’s largest basin…”
Etc.…many more changes to wording and grammar are still needed.
Lns 52-52: Remove section about rainforest—is it needed?
Lns 55-56: This is not a complete sentence.
Lns 52-67: This section can probably be reduced in length
Ln 62: What is meant by “less agreement” Does it refer to the previous sentence?
Lns 73-76: Unclear, needs to be re-written.
Lns 73-81: Not clear how this information is being used to motivate the specific study. The overall structure of the introduction remains unclear and need to be tightened up to clearly identify a research gap that motivates the study’s objectives.
Ln 82: Introduction of socio-economic context is abrupt and unsupported. Electricity is not the only economic output – in particular fisheries are also very important and in conflict with hydropower. Moreover, there are many other facets of the “social” side of socio-economics.
Lns 87-89: So what was found through these assessments? Also, needs a rewrite: “Worldwide, the decrease of sediment transport downstream of dams is a major impact that has been ASSESSED [28–31]. However, some studies ASSESSING this failed to ASSESS the natural and anthropogenic factors…”
Lns 93-94: What does it mean to have the largest index? And how does that fact inform this study?
Ln 112: Remove “only in”
Ln 164: What is the “cross-section of suspended sediment…”?
Ln 167: main stem, not main stream
Lns 198-200: This is hard to interpret. Why the “However”?
Ln 213: High INTERANNUAL variability.
Lns 217-232: This paragraph is very difficult to follow as written. Remove “registered” from all sentences. Also, the authors refer to discharge and FSC in specific months and years, but point the reader to figure 2, which shows their distribution, which is confusing. It is not clear why the authors quantify specific months and years in terms of others and why they choose specific time periods over which to do so, e.g.,
“A similar dilution process was also observed in CE during March 2014, with an estimated FSC of 450 mg L−1, which is 75% less than the average (1800 mg L−1) of values from the March of eleven previous years and 68% less than the values of the March in the three succeeding years (1400 mg L−1) (Figure 2a).”
All of the writing needs to be tightened up and have a specific topic or goal for each paragraph.
Ln 236-252: This paragraph is even more difficult to follow. For example, the first two sentences are totally incomprehensible to me, though I think they refer to the flow exceedance curve: “At PV it was observed that during the 2005 drought the discharge exceeded half of that year’s volume by 12,850 m3 s−1. This is 18% less than the 50% exceeded estimated during this research period (15,700 m3 s−1). The 2014 flood generated a discharge exceeding half of that year’s 26,000 m3 s−1. This is 65% more than what was estimated during this research period.”
The third sentence makes sense: “The highest mean monthly discharge observed at PV was during March 2014 (56,000 m3 s−1).
Then I lose the meaning again: “When compare the previous event in 2015 the discharge exceeded half of that year’s 26,000 m3 s−1 was similar to 2014” …. “This is the result from the differences over the monthly discharges during the research period. Though, 2015, presented larger monthly mean discharges from December 2014 (i.e., 17% more than December’s mean), that extended to August, 2015, (i.e., 100% more than Augusts’ mean).”
All of the writing needs to be tightened up and have a specific topic or goal for each paragraph.
Ln 265: Text says that r = 0.2, but figure shows R^2 = .02; these don’t agree (0.2^2 = 0.04)
Lns 281-287: Here and elsewhere, the manuscript would benefit from separating results from discussion. This is up to the authors and editors, but mixing results and discussion further reduces the clarity of the very good analysis and results.
Lns 302-303: What direction is the trend? Is this shown in Fig. 5?
Lns 302-321: I think it would be very helpful to have a table that summarized the results of all the time series results. So, the table would summarize the nonparametric Pettitt p-values and change point years, Buishand test and Von Neumann ratio results (not sure what these are), and Mann-Kendall slopes and p-values for each response variable (Q and FSC) at each of the three stations. As written it is very hard to synthesize the overall findings.
Ln 340: Need to highlight what it means that Q was not changing even though FSC was.
Lns 349-359: Why add hypotheses at the END of the manuscript? If they were all untestable, I suggest removing them as hypotheses, but instead just calling them possible explanations and describing the methods/analyses that would be required to test them (in future work).
Author Response
Thank you for the second revision of our manuscript and the helpful comments. Please find below our responses. The Water English Specialist editors revised the new version of the manuscript.

Reviewer 2 Report
From the former version, the manuscript has been modified according to this reviewer's suggestions, so I recommend its publication in present form.
Author Response
Thank you for the second revision of our manuscript and your approval. As suggested the Water English Specialist editors revised the new version of the manuscript.